# Laboratory infrared spectra and fragmentation chemistry of sulfur allotropes

Piero Ferrari [1] ✉, Giel Berden [1], Britta Redlich[1], Laurens B. F. M. Waters[2] & Joost M. Bakker [1]

Sulfur is one of six life-essential elements, but its path from interstellar clouds to planets and their atmospheres is not well known. Astronomical observations in dense clouds have so far been able to trace only 1 percent of cosmic sulfur, in the form of gas phase molecules and volatile ices, with the missing sulfur expected to be locked in a currently unidentified form. The high sulfur abundances inferred in icy and rocky solar system bodies indicate that an efficient pathway must exist from volatile atomic sulfur in the diffuse interstellar medium to some form of refractory sulfur. One hypothesis is the formation of sulfur allotropes, particularly of the stable $S_8$. However, experimental information about sulfur allotropes under astrochemically relevant conditions, needed to constrain their abundance, is lacking. Here, we report the laboratory far-infrared spectra of sulfur allotropes and examine their fragmentation pathways. The spectra, including that of cold, isolated $S_8$ with three bands at 53.5, 41.3 and 21.1 μm, form a benchmark for computational modelling, which show a near-perfect match with the experiments. The experimental fragmentation pathways of sulfur allotropes, key information for astrochemical formation/destruction models, evidence a facile fragmentation of $S_8$. These findings suggest the presence of sulfur allotropes distributions in interstellar space or in the atmosphere of planets, dependent on the environmental conditions.

Sulfur is the tenth most abundant element in the universe, with a cosmic [S/H] ratio of $1.4 \cdot 10^{-5}$, which together with H, C, O, N and P, is regarded as one of the essential elements for life[1]. Moreover, sulfur is found in nature in a rich variety of allotropic forms, only surpassed in number by those of carbon[2].

However, the pathway of sulfur from interstellar gas clouds to planets is a long-standing question[3–6]. While in low-density diffuse interstellar environments the observed atomic sulfur, expressed in its abundance ratio to atomic hydrogen, is close to the cosmic value[7,8], in dense molecular clouds, star-forming regions and in planet forming disks that surround young stars, the gas phase sulfur concentration (measured as the sum of all atomic sulfur and sulfur atoms contained in gas phase molecules) is strongly depleted, by up to two orders of magnitude[9,10]. Volatile sulfur ice reservoirs such as OCS and $SO_2$ can account for at most a few per cent of the missing sulfur[11]. In contrast, icy and rocky solar system bodies show abundant sulfur. Samples with a roughly solar [S/O] ratio were recovered by the Rosetta mission to the comet 67 P/Churyumov-Gerasimenko[12], with about 80 per cent of sulfur in some refractory form. Refractory sulfur is also abundant in primitive meteorites of the CI class[13], where it is found as sulfides and sulfates[14]. The sample return mission to the CI asteroid Ryugu[15] has revealed similar high sulfur abundances. Recently, sulfur has for the first time been detected in the atmospheres of gas giant exoplanets[16,17]. These observations show that sulfur is efficiently converted from atomic to refractory reservoirs, even in icy objects like comets.

[1]Radboud University, Institute for Molecules and Materials, FELIX Laboratory, Nijmegen, The Netherlands. [2]Department of Astrophysics, IMAPP, Radboud University, Nijmegen, The Netherlands. ✉e-mail: piero.ferrariramirez@ru.nl

Sulfur allotropes have been proposed as an important sulfur reservoir in molecular clouds[18], formed, for example, upon UV irradiation of $H_2S$ ices[19], or electron irradiation of $H_2S$ and $SO_2$ ices[20]. In the latter laboratory study, an apparent depletion of accountable sulfur budget was observed, attributed to the possible formation of sulfur allotropes. Indeed, $S_2$, $S_3$ and $S_4$ were identified at trace abundances in 67 P/Churyumov-Gerasimenko[12], measured with an instrument sensitive towards the small sulfur allotropes. Moreover, $S_8$ was detected in Ryugu samples[21]. Sulfur allotropes may also play an important role in providing UV opacity in the Venusian atmosphere[22], and in Venus-type exoplanets[23]. Density functional theory (DFT) calculations have suggested that among the $S_N$ allotropes, octasulfur $S_8$ is especially stable[24], with astrochemical models highlighting its relevance in the dynamics of sulfur in the interstellar medium (ISM). Sulfur allotropes may therefore be the missing link between the diffuse interstellar medium and solar system parent bodies. Nevertheless, a lack of relevant spectroscopic and thermodynamic information on sulfur allotropes has prevented a proper investigation of this hypothesis as part of the solution of the sulfur puzzle. While sulfur allotropes have been the subject of active investigation[25–27], spectroscopic information under astrochemically relevant conditions are scarce, with one example being the rotational spectra of $S_3$ and $S_4$[28].

Here, we report the far-infrared spectrum of isolated neutral $S_8$, under the cold and isolated conditions of a molecular beam. In addition, infrared spectra of the ions $S_4^+$ and $S_4^-$ are recorded in a room temperature ion trap. The experimental spectra of the investigated species show a remarkably good agreement with computational modelling, enabling us to predict lower abundance limits for their astronomical detection using the James Webb Space Telescope (JWST)[29]. These results can allow the targeted observational search of sulfur allotropes. The spectral information is complemented by information on fragmentation energetics and pathways providing necessary input for astrochemical modeling of the sulfur inventory in dense molecular clouds and star-forming regions.

## Results

### Far-infrared spectroscopy of the $S_8$ allotrope

A cold and diluted molecular beam of neutral sulfur allotropes is generated from sulfur powder. A mass spectrum of the typically generated distribution (Supplementary Fig. 1) recorded after ionization with 118 nm laser light (10.5 eV/photon) shows allotropes from $S_2$ to $S_8$. The sulfur atom is not observed, possibly due to its high ionization energy (10.4 eV), barely below the energy of the ionization light. The possibility that atomic S is present in the molecular beam, however, cannot be excluded. Moreover, we note that in the past sulfur allotropes larger than $S_8$ have also been suggested to be stable, but that these were not observed here[30]. The present experimental distribution is dominated by the octasulfur allotrope $S_8$, likely because the precursor sulfur powder may be composed largely of α-sulfur, formed by stacks of $S_8$ units. Supplementary Fig. 2 provides the infrared spectrum of solid α-sulfur, supporting this idea. We note that fragmentation of $S_8$ induced by the photoionization is unlikely, given that the sum of ionization and fragmentation energies, discussed later in this contribution, is higher than 10.5 eV.

The far-infrared spectrum of neutral $S_8$ in the 150–600 $cm^{-1}$ (67–17 μm) spectral range is recorded via infrared photodissociation spectroscopy, utilizing the free-electron laser FELIX (Nijmegen, The Netherlands)[31]. The spectrum, presented in Fig. 1a, is composed by registering fragmentation of $S_8$ into both $S_5$ and $S_6$. It shows three clear bands, centered at 187, 242 and 474 $cm^{-1}$ (53.5, 41.3 and 21.1 μm).

Because this experiment relies on the absorption of more than one photon to achieve fragmentation (with the condition that the energy carried by the photons exceeds the fragmentation energy) and we cannot establish how many photons are absorbed per infrared pulse, the experiment does not allow to obtain absolute

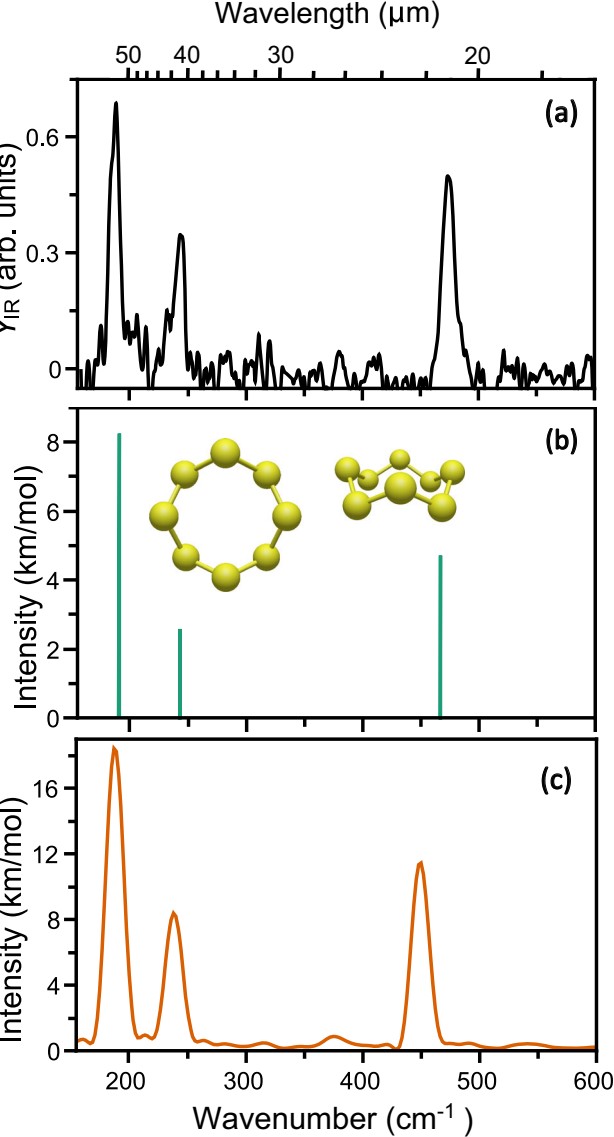

**Fig. 1 | Infrared spectrum of neutral $S_8$. a** Experimental far-infrared spectrum of gas-phase neutral $S_8$. **b** Harmonic vibrational modes of $S_8$, calculated by density functional theory using the geometry shown as inset, where two views of it are presented. **c** Far-infrared spectrum of $S_8$ computed using molecular dynamics simulations at 50 K. Source data are provided as a Source Data file.

absorption cross-sections. Nevertheless, the relative absorption cross-sections are crucial to benchmark computational methods. Previous computational studies exploring the potential energy surface of $S_8$ reported a crown-shaped ring geometry with singlet spin multiplicity as the putative ground state[32]. Here, we show the harmonic vibrational frequencies of $S_8$ for such a geometry (Fig. 1b). The computed vibrational bands at 191, 243 and 467 $cm^{-1}$ agree remarkably well with the experimental spectrum, both in line positions and in relative intensities. Because of the high $D_{4d}$ symmetry of $S_8$, the modes at 191 and 467 $cm^{-1}$ are doubly degenerate, whereas the mode at 241 $cm^{-1}$ is non-degenerate. Normal mode vectors are depicted in Supplementary Fig. 3. This high symmetry means that $S_8$ has no permanent dipole moment, rendering it invisible to microwave spectroscopy.

The spectral bandwidths of the three observed bands (full-width at half-maximum, FWHM) are close to 10 $cm^{-1}$, and therefore larger than the IR laser bandwidth of FELIX of 1–2 $cm^{-1}$ at these wavelengths. The observed bandwidths could result from a broadening effect innate

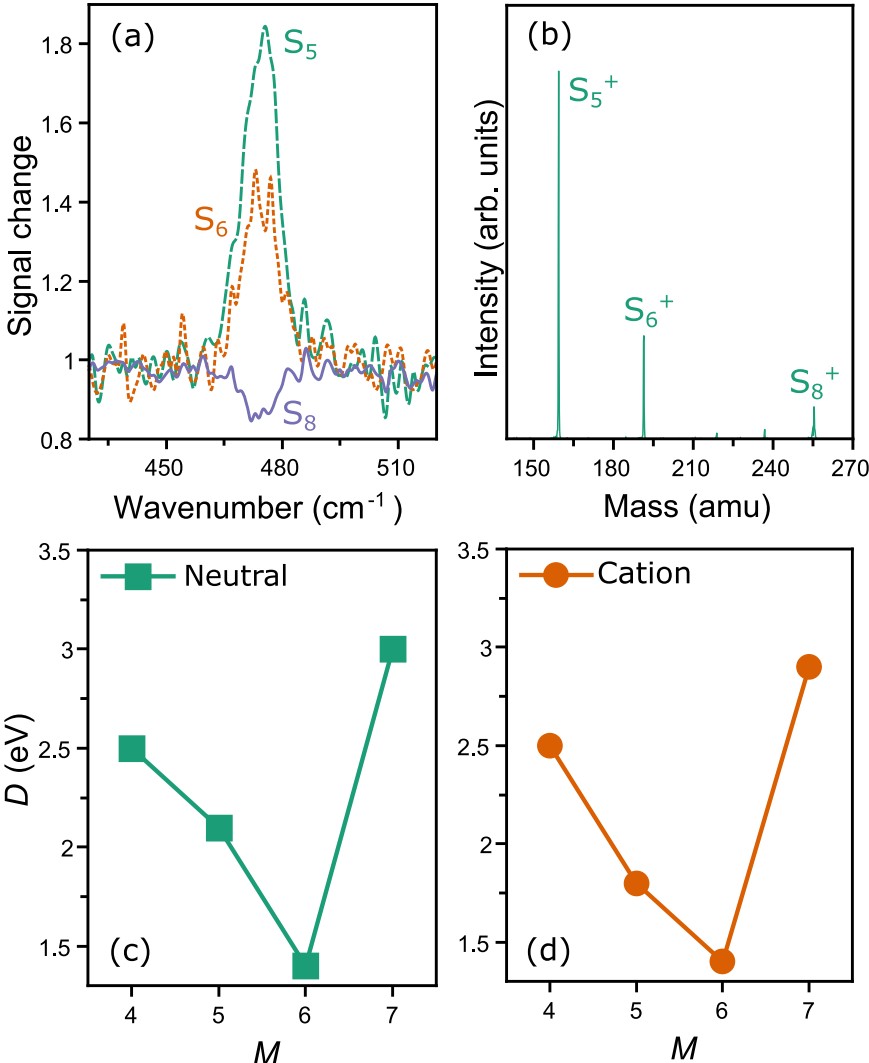

**Fig. 2 | Fragmentation of sulfur clusters. a** Wavenumber dependence of the laser-induced reduction of $S_8$ signal compared with the signal increase detected for $S_5$ and $S_6$. **b** Fragmentation products after the collision induced dissociation of isolated $S_8^+$ clusters. **c, d** Density functional theory calculated fragmentation energies of $S_8$ and $S_8^+$ fragmenting into $S_M$ and $S_{8-M}$, or $S_M^+$ and $S_{8-M}$ species, respectively. Source data are provided as a Source Data file.

to the requirement to absorb > 20 IR photons to reach the fragmentation threshold. To test an alternative scenario of dynamical broadening at finite-temperatures, an ab-initio Born–Oppenheimer molecular dynamics (BOMD) simulation was performed. From such simulations, an IR spectrum can be obtained that includes the intrinsic vibrational linewidths, and directly comprises anharmonic effects. The temperature in previous molecular beam studies in the same experimental instrument was shown to range from 40 to 50 K[33], so we took 50 K as an upper limit here. The result of the 5 ps, 0.5 fs step simulation is shown in Fig. 1c. This simulated spectrum also reproduces the three main bands of $S_8$, with only a slightly poorer agreement in band positions than the density functional theory calculation. The dynamics simulations show that the width of these bands is intrinsically larger than the bandwidth of FELIX, as a consequence of the shape fluctuations of $S_8$ at 50 K. They simultaneously demonstrate that at 50 K, $S_8$ is stable, with only small shape fluctuations of the ground-state geometry, excluding fragmentation or isomeric changes. Finally, smaller features between the main bands are predicted, providing a possible agreement with weaker modes detected close to the experimental noise level. These intrinsic widths of the bands will complicate astronomical observations of free $S_8$, as discussed later.

## Destruction pathways of neutral sulfur allotropes

In addition to the measurement of the far-IR spectrum of $S_8$, the experiments allow a determination of its destruction pathways induced by IR absorption. Because the infrared-induced fragmentation mechanism is statistical in nature, fragmentation follows the lowest-energy pathways, and is likely the same which follows after the absorption of UV or visible photons. Observed fragmentation patterns are therefore crucial for astrochemical modelling of molecular abundances, so far relying on general assumptions or computations[30,34]. Figure 2a presents the wavelength-dependent depletion and appearance (the mass-spectral intensity ratios with and without IR laser light) of neutral $S_8$, $S_6$ and $S_5$. As seen from the figure, a decrease in $S_8$ signal (corresponding to laser induced fragmentation) coincides with an increase in $S_6$ and $S_5$ intensities. Such increases are not observed for the $S_7$ and $S_4$ channels. Therefore, it is shown experimentally that $S_8$ fragments via a competition of the channels $S_8 \rightarrow S_5 + S_3$ and $S_8 \rightarrow S_6 + S_2$. We note that a clear signal increase for $S_2$ and $S_3$ is not observed, possibly because of the larger recoil energy of smaller fragments, making them more difficult to detect.

To rationalize the observed fragmentation channels for $S_8$, thermodynamic fragmentation energies for the possible $S_8 \rightarrow S_M + S_{8-M}$

($M$ = 4–7) pathways (Fig. 2c) are calculated. The lowest fragmentation energies are calculated for the channels leading to $S_5$ and $S_6$ products. Although both are observed in the experiment, we note that the experiment favors $S_5$ formation, whereas the fragmentation energy $D$ calculated for forming $S_6$ is lower. The reliability of the density functional theory calculated fragmentation energies are confirmed by single-point coupled-cluster (CCSD(T)) calculations (Supplementary Fig. S4). The seemingly conflicting findings between experiment and computations can be evaluated further by not only taking thermodynamic, but also kinetic factors in the fragmentation reaction into account. During fragmentation, the system will encounter energy barriers, for instance when S–S bonds are broken. The values in Fig. 2c therefore correspond to lower limits of the energy needed to fragment $S_8$. This is explored by calculating the lowest-energy pathway along the potential energy surface describing the $S_8 \rightarrow S_6 + S_2$ fragmentation reaction. Supplementary Fig. S5 shows two energy barriers at 2.2 eV above the energy of $S_8$, corresponding to the breaking of the two S-S bonds, placing a higher, kinetic energy limit for fragmentation. Further, it should be considered that fragmentation rates are not only dependent on energetics, but also on the associated entropy change favoring pathways towards trimers over dimer loss[35]. Nevertheless, the key observations are the experimentally determined fragmentation pathways $S_8 \rightarrow S_5 + S_3$ and $S_8 \rightarrow S_6 + S_2$.

We note that the presence of sulfur in the atmospheres of gas giant exoplanets is used as evidence for photochemistry in an atmosphere that is enhanced in metals. This assumed tracer role of sulfur depends strongly on a non-volatile nature of the sulfur reservoir in planet forming disks. Our study reveals the fragmentation pathways of $S_8$, which lead to more volatile sulfur allotropes, and hence would increase the volatility of sulfur in disks. If $S_8$ is indeed a major reservoir of sulfur in planet forming disks, this would thus imply that sulfur is less reliable as a tracer of the metal content in gas giant atmospheres.

## Destruction pathways of charged sulfur allotropes

Because local surroundings determine the charge state of interstellar species, with both neutrals and ions being currently identified in the ISM[36], we also investigated negatively and positively charged sulfur allotropes, using a room temperature quadrupole ion trap[37]. Ions were formed through sublimation at 250 °C and ionization in a plasma corona discharge, before being guided to the He filled ion trap. As shown in Supplementary Fig. S6, this preparation method leads primarily to $S_8^+$ in the cationic charge state. For anions, only $S_4^-$ is observed.

Experimental information about the destruction pathways of ionic sulfur allotropes is obtained via collision induced dissociation (CID)[38], where a specific allotrope is isolated in the ion trap and the products of fragmentation induced by collisions with He gas are characterized. Isolating and fragmenting the $S_8^+$ allotrope leads to the formation of $S_5^+$ and $S_6^+$ products, in line with previous findings[39]. This is shown in the mass spectrum of Fig. 2b, where upon collision induced dissociation the intensity of the isolated $S_8^+$ decreases, with a concomitant increase in the $S_5^+$ and $S_6^+$ channels. Therefore, $S_8^+$ follows the $S_8^+ \rightarrow S_5^+ + S_3$ and $S_8^+ \rightarrow S_6^+ + S_2$ destruction pathways, like neutral $S_8$. Here as well, calculations of fragmentation energies (Fig. 2d) show the lowest values for the channels forming cationic $S_5^+$ and $S_6^+$ products, in line with the experimental observations, although the $S_5^+$ channel is again not the thermodynamically favored.

Following the fragmentation of $S_8^+$, the $S_6^+$ and $S_5^+$ products can be further isolated, allowing also characterization of their fragmentation pathways. $S_6^+$ was found to have a unique $S_6^+ \rightarrow S_4^+ + S_2$ destruction pathway, whereas for $S_5^+$ there is a competition between the $S_5^+ \rightarrow S_3^+ + S_2$ and $S_5^+ \rightarrow S_2^+ + S_3$ channels. Finally, $S_4^+$ and $S_3^+$ fragment following $S_4^+ \rightarrow S_2^+ + S_2$ and $S_3^+ \rightarrow S_2^+ + S$. A summary of calculated fragmentation energies for the different cationic sulfur clusters is presented in Supplementary Fig. 7.

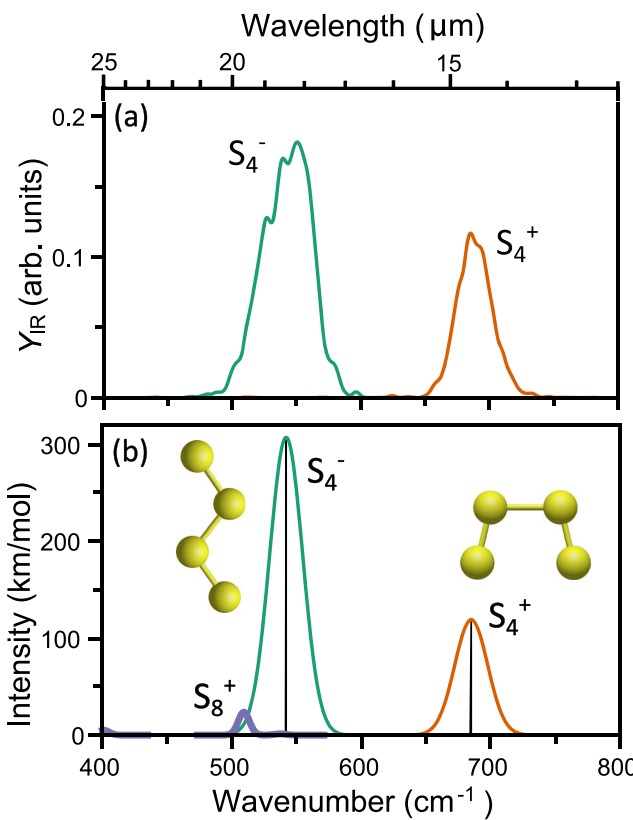

**Fig. 3 | Infrared spectra of charged allotropes. a** Experimental far-infrared spectra of $S_4^+$ and $S_4^-$, measured in a room temperature ion trap. **b** Density functional theory calculated harmonic vibrational modes of $S_4^+$ and $S_4^-$, using the shown geometries. Transitions are broadened using Gaussian line shapes, broader than the laser spectral profile, to simulate broadening effects in the experiment. For comparison, the calculated vibrational spectrum of $S_8^+$ is also shown. Source data are provided as a Source Data file.

## Far-infrared spectroscopy of ionic $S_4^+$ and $S_4^-$ allotropes

IR induced fragmentation of $S_8^+$ proved unsuccessful, potentially due to a combination of a high fragmentation threshold, medium strength IR cross sections, large heat capacity, and importantly, the presence of He in the ion trap, acting as a heat bath that prevents reaching the required internal energy for fragmentation.

In contrast, for the dominant anionic $S_4^-$ allotrope, as well as for the generated $S_4^+$ fragment, we successfully recorded an IR spectrum. For $S_4^-$ (Fig. 3a) only a single mode is detected in the 400–800 cm$^{-1}$ (12.5–25 μm) spectral range covered in the ion trap experiments, centered at 544 cm$^{-1}$ (18.4 μm). Concomitant calculations yielded a single $S_4^-$ harmonic vibrational mode at 542 cm$^{-1}$ (Fig. 3b), showing a near-perfect match with the experiment. The IR spectrum of $S_4^+$ (Fig. 3a) has, analogous to the $S_4^-$ anion, only a single IR band within the 400–800 cm$^{-1}$ range, centered at 688 cm$^{-1}$ (14.5 μm). A calculation again finds an almost perfect agreement with the experiment, predicting a single vibration at 685 cm$^{-1}$ (Fig. 3b). For comparison, the calculated harmonic vibrational spectrum of $S_8^+$ shows a much lower IR activity than $S_4^+$ and $S_4^-$, a potential reason for the failure to observe infrared-induced fragmentation of $S_8^+$.

## IR absorption cross sections of sulfur allotropes and comparison to comet 67 P/Churyumov-Gerasimenko

Our experiments reveal IR absorption bands for $S_N$ allotropes concentrated in the 15–22 μm range, with a trend of decreasing wavelength for smaller $N$. Interestingly, this wavelength range coincides with the prominent absorption detected in irradiated laboratory samples that

are proposed to be representative of the refractory sulfur-containing organics detected in the comet 67 P/Churyumov-Gerasimenko[40]. In particular, an absorption peak near 21 μm is close to the 21.1 μm band of $S_8$. Given that $S_2$ and $S_3$ desorption products were detected in the warm-up of the laboratory ices, we suggest that the organic residue also contains sulfur allotropes, including $S_8$. Because of the relatively low fragmentation energies of only a few eV of most allotropes, the detection of $S_2$ and $S_3$ in irradiated sulfur bearing ices may well result from larger sulfur allotropes. We note that $S_2$ and $S_3$ were detected both in comet and in the laboratory residue[40], and that $S_2$ is ubiquitous in cometary comae[41]. This further supports the role of sulfur allotropes as a major sink of sulfur. However, we conjecture that this should be in the form of a distribution, with octasulfur $S_8$ as most stable representative, but subject to fragmentation by external fields. Our results support the notion that the presence of $S_3$ and $S_4$ and a significant fraction of the $S_2$ in the comet 67 P/Churyumov-Gerasimenko[12] are due to fragmentation of $S_8$.

Based on our results we can estimate the requirements for detecting sulfur allotropes in the ISM using the JWST MIRI instrument. For neutral $S_8$ (Fig. 1a), vibrational modes are experimentally seen at 53.5, 41.3 and 21.1 μm. Of these, only the 21.1 μm band lies within the accessible spectral range[42] and therefore, could be targeted for detecting $S_8$ in space. A key question, however, is the absolute cross section of such a band. Given the excellent agreement between the experimental infrared spectrum and the computed vibrational frequencies and relative intensities presented here, we propose the computational results as a reliable source for deducing the cross sections. Under the assumption of a rigid $S_8$ structure, the instrument resolution limited absorption cross-sections for the 21.1 μm band increase from $2.5 \cdot 10^{-19}$ to $8.0 \cdot 10^{-19}$ cm$^2$ when reducing the temperature from 40 to 1 K due to the reduction in population of excited rotational states. Of course, a flexible nature of $S_8$, as inferred from the molecular dynamics simulations, lowers these values. We have simulated IR profiles of JWST/MIRI observations, indicating that the direct detection of cold gas phase $S_8$ in molecular clouds is challenging, based on the required S/N ratio of the 21.1 μm band in the IR profiles, under the assumption that a significant fraction of the sulfur content goes into $S_8$. More details are provided in the Supplementary Figs. 8 and 9. Furthermore, we note that other allotropes have much larger cross sections than $S_8$, such as $S_8^+$ with $\sigma = 2.6 \cdot 10^{-18}$ cm$^2$ for the predicted band at 19.7 μm, $S_4$ with $\sigma = 1.4 \cdot 10^{-17}$ cm$^2$ for the predicted band at 15.4 μm, $S_4^+$ with $\sigma = 1.4 \cdot 10^{-17}$ cm$^2$ for the measured band at 14.5 μm, and $S_4^-$ with $\sigma = 3.7 \cdot 10^{-17}$ cm$^2$ for the measured band at 18.4 μm (values calculated at 5 K). A list of cross sections calculated for different sulfur allotropes is presented in Supplementary Table 1. The possibility of detecting other allotropes of course still depends on their abundance in the ISM.

## Final remarks

The IR spectral properties and photostability of neutral $S_8$ in the cold and isolated conditions of a molecular beam were investigated, underlying its possible presence in cold interstellar environments. Its far-infrared signature reveals a characteristic mode at 21.1 μm, within reach by JWST, as well as clear bands at 41.3 and 53.5 μm. Computations of the IR spectrum of $S_8$, as well as those of $S_4^+$ and $S_4^-$, measured in a room-temperature ion trap, reveal a near-perfect agreement with experimental results. These benchmark results for the calculations lend credibility to future computational work on other sulfur allotropes. For both neutral $S_8$ and its cationic and anionic counterparts, the destruction pathways demonstrate a pronounced stability consistent with calculated thermodynamic fragmentation energies >1.5 eV that with inclusion of kinetic barriers likely exceed 2 eV. Our data allow direct tests of the presence of sulfur allotropes in space and in laboratory samples studied with infrared spectroscopy. While the direct detection of $S_8$ in space is challenging, the presence of $S_2$, $S_3$ and $S_4$ are signposts of the fragmentation of $S_8$, strengthening the evidence

that $S_8$ is a major sink of sulfur in space. The measured IR spectra, together with the observed fragmentation chemistry, provide a key piece for solving the long-standing Sulfur Depletion Puzzle[43].

## Methods

### Molecular beam experiments on neutral clusters

Neutral sulfur clusters, $S_N$ ($N = 2–8$), are formed by laser desorption of sulfur powder (Sigma-Aldrich; 99.98%) mixed with carbon black in a nearly 1:1 ratio. The mixture is pressed on a graphite sample bar that is excited by a mildly focused Nd:YAG laser (1064 nm, 10 Hz, 1 mJ/pulse), while being translated at a fixed speed. The desorbed sulfur is entrained in the expansion of a pulse of argon, released from a pulsed valve operated at a backing pressure of 2 bar, forming a supersonic beam. Collisional cooling with argon thermalizes the entrained sulfur species to typical temperatures of 50 K under these conditions. Support of this assumption is given by comparisons with molecular dynamics simulations. The formed molecular beam is collimated by a 2 mm skimmer before it enters the extraction region of a reflectron time-of-flight mass spectrometer, with a typical mass resolution of $m/\Delta m = 2500$ at 256 amu ($S_8$ mass). The neutral sulfur clusters are probed after ionization with 118 nm (10.5 eV) laser light, generated by frequency tripling the third harmonic of a Nd:YAG laser (355 nm, 10 Hz, 18 mJ/pulse) in a Xe/Ar cell (20 mbar of Xe in 150 mbar of Ar). A scheme of the experimental setup is presented in ref. 44.

The infrared spectrum of $S_8$ is measured via IR photofragmentation spectroscopy. For this, the molecular beam is illuminated by IR light produced by the free-electron laser FELIX (Nijmegen, The Netherlands). The laser and molecular beams are counter propagating, and IR illumination with a single FELIX pulse (7 μs, 50-100 mJ) takes place roughly 300 μs before ionization. Upon resonant excitation with a vibrational mode of $S_8$, multiple-photon excitation heats the cluster until it overcomes the fragmentation threshold, leading to fragmentation. By running FELIX at 5 Hz, half the molecular beam repetition rate, successive mass spectra with and without IR light are recorded allowing for correction of long-term source fluctuations. Resonant absorption is identified from mass-spectral signal depletion of $S_8$, in addition to the signal increase of $S_5$ and $S_6$. This also allows to identify the fragmentation channels of $S_8$. The infrared spectrum of $S_8$ is expressed in the IR yield

$$Y_{IR}(\nu) = -\ln\left(\frac{B(\nu)}{B_0}\right) / E(\nu) \qquad (1)$$

with $B(\nu)$ and $B_0$ the branching ratio with FELIX excitation at frequency $\nu$ and without it, respectively. $E(\nu)$ corresponds to the IR laser pulse energy. $B$ is defined as:

$$B = \frac{I_8}{I_5 + I_6 + I_8} \qquad (2)$$

with $I_N$ the signal intensity for allotrope $S_N$. To obtain the spectra, FELIX is scanned in the 150–600 cm$^{-1}$ (67–17 μm) spectral range, in steps of 1 cm$^{-1}$. The FELIX spectral bandwidth is optimized to have at maximum a standard deviation of 0.5% of the central wavelength.

### Ion trap experiments on cationic clusters

The infrared spectra of $S_4^+$ and $S_4^-$ ions are measured at room temperature on a Bruker amaZon quadrupole ion trap, modified to have optical access to the trapped ions. Details of the experimental setup can be found in ref. 37, which has been used in the past to record infrared spectra of astrochemically relevant molecules, such as fullerene-derivatives, showing good correspondence with astronomical observation[45]. Sulfur powder is used as precursor in an atmospheric pressure ionization source equipped with a direct insertion probe, where the material is placed at the tip of a glass tube heated to 250 °C.

After sublimation, the material is ionized in the plasma of a corona discharge, using a potential difference between the end plate and the capillary of 4000 V and a corona current of 6000 nA. This process leads to the formation of $S_8^+$ cations or $S_4^-$ anions, which enter the radio-frequency ion trap where they can be mass-isolated. The IR spectrum of $S_4^-$ is obtained by irradiating the trapped ions with a single FELIX pulse after which the fragmentation yield is derived from the mass spectrum by monitoring the intensity of $S_4^-$ and $S_2^-$ anions. $S_8^+$ is dissociated by collisions with He gas via collision-induced dissociation (CID). Mass-isolation of $S_6^+$ followed by CID produced $S_4^+$, which was mass-isolated and irradiated with 5 pulses of FELIX. In this case, the IR induced fragment ion was $S_2^+$. The IR induced fragmentation yield was linearly corrected for the IR laser pulse energy[46].

### Density functional theory calculations

The vibrational modes of sulfur allotropes are obtained from density functional theory (DFT) calculations, performed with the ORCA 5.03 software package[47]. The Def2-TZVPP basis set is employed, in addition to dispersion corrections to total energies via the D3BJ method. Calculations are performed with the "verytight" convergency criteria for the SCF cycles and the geometry optimizations, as implemented in ORCA. An initial benchmark analysis of different exchange-correlation functionals was conducted, with harmonic vibrational modes compared with the measured infrared spectrum of $S_8$. This comparison is presented in Supplementary Fig. 10 and involves calculations employing the PBE (GGA), TPSS (meta-GGA), PBE0 and B3LYP (hybrids), CAM-B3LYP (long-range separated hybrid) and B2PLYP (double hybrid) functionals. While all functionals yield harmonic infrared spectra closely resembling the experimental spectra, B3LYP is selected for all computations presented here, given its better match with the relative intensities in the experiment. The geometries employed for the calculations are obtained from literature[48], and re-optimized here (Supplementary Fig. 11). Vertical ionization energies ($IE_N$) of the neutral $S_N$ clusters are also computed, using Eq. (3), with $E$ the zero-point corrected energy of the cluster within brackets. In this case, single-point calculations for $S_N^+$ are performed on the optimized geometry of the corresponding $S_N$ cluster, to account for a vertical transition.

$$IE_N = E(S_N) - E(S_N^+) \tag{3}$$

The dissociation energy ($D$) of allotrope $S_N$ or $S_N^+$, fragmenting into $S_M + S_{N-M}$ or $S_M^+ + S_{N-M}$, respectively, is computed via Eqs. (4) or (5),

$$D = E(S_N) - E(S_M) - E(S_{N-M}) \tag{4}$$

$$D = E(S_N^+) - E(S_M^+) - E(S_{N-M}) \tag{5}$$

The computed vibrational spectra of all $S_N$ and $S_N^+$ allotropes are presented in Supplementary Figs. 12 and 13.

### Ab initio Born–Oppenheimer molecular dynamics simulations

The infrared spectrum of neutral $S_8$ is also constructed from a dynamical picture using Ab initio Born–Oppenheimer molecular dynamics (BOMD) simulations. In this case, the CP2K 6.1 software package[49] is employed, using the B3LYP functional and the TZV2P-GTH basis set, including D3BJ dispersion corrections. A density cutoff of 400 Ry is fixed, and the simulations run in a vacuum periodic orthorhombic box of 25 Å length. The dynamics starts with the $S_8$ geometry determined by DFT, after which a total time of 5 ps is simulated, in steps of 0.5 fs. A canonical (NVT) ensemble is employed, with a temperature of 50 K, constrained through a velocity rescaling thermostat. Molecular dipole moments are evaluated from the maximally localized Wannier function centers, and the infrared spectrum is calculated from these coordinates using the TRAVIS software package[50].

## Data availability

All the data supporting this study are available in the article and associated Supplementary Information. Source data are provided with this paper.

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

## Acknowledgements

We gratefully acknowledge the Nederlandse Organisatie voor Wetenschappelijk Onderzoek (NWO) for the support of the FELIX Laboratory and thank the FELIX staff. We thank Dr. Nils Deßmann for support with Fourier-transform infrared (FTIR) spectroscopy measurements.

## Author contributions

P.F., L.B.F.M.W. and J.M.B. conceived the project. P.F. performed the molecular beam experiments and the computations. G.B. and P.F. performed the ion trap experiments. P.F. and J.M.B. conducted the data analysis. B.R. secured the funding. All authors interpreted the data and co-wrote the manuscript.

## Competing interests
The authors declare no competing interests.
