## [Peer Review File · Nature Communications]

Laboratory infrared spectra and fragmentation chemistry of sulfur allotropesREVIEWER COMMENTS

Reviewer #1 (Remarks to the Author):

This paper reports some new and important results on the spectroscopy of sulfur allotropes and examine their fragmentation pathways. The role of sulfur in astrochemistry is an important one with a major problem being the lack of observations of sulfur in many astronomical regions. The determination of the forms in which sulfur may be found is therefore subject of debate and laboratory research, this is the 'sulfur depletion puzzle.' The presented paper focuses on sulfur being tied up in several allotropes, in particular S₈ which may be fragmented to form S_x where x=2,3,4 in both anionic and cationic forms. The paper presents experimental IR spectral measurements and the photodissociation of S₈ revealing a characteristic mode at 21.10 μm, which can be detected by the JWST MIRI instrument, as well as other bands at 41.32 and 53.48 μm. The paper used these results to suggest the presence of S₂ S₃ and S₄ in the comet 67P/Churyumov Gerasimenko arising from fragmentation of S₈. The experimental work is supported by DFT calculations. The methodology is well established and the results well defined and discussed however the Authors should address the following points.

1. In the second paragraph of the introduction, the authors state that "volatile sulfur ice reservoirs such as OCS and H₂S can account for at most a few percent of the missing sulfur." However H₂S has yet to be detected in interstellar icy mantles (see, e.g., McClure et al. *Nature Astronomy*, 7, 431, 2023). I believe the authors may have meant to write SO₂ instead of H₂S.
2. Although the authors describe the logic behind the conversion of volatile (i.e., atomic or refractory) sulfur to refractory (i.e., allotropic or mineral sulfur), they do not go into any detail about possible methods by which this may occur. A few short sentences on this should be added. I refer them to the papers of Cazaux et al. (*Astronomy and Astrophysics*, 657, A100, 2022) who showed that the photo-processing of H₂S ice leads to the formation of refractory sulfur allotropes, while Mifsud et al. (*Frontiers in Chemistry*, 10, 1003163, 2022) showed that electron irradiation of SO₂ or H₂S generates an apparent "depletion" of observable sulphur. Moreover, Coulomb-enhanced freeze out of sulphur ions onto dust grains (see Ruffle et al. *Monthly Notices of the Royal Astronomical Society*, 306, 691, 1999; Umebayashi and Nakano, *Publications of the Astronomical Society of Japan*, 32, 405, 1980) may provide the opportunity for a gradual atom-by-atom build-up of sulphur residues.
3. In the penultimate paragraph of the introduction, the authors state that "To date, the only spectroscopic information on sulfur clusters are the rotational spectra of S₃ and S₄". This is rather misleading since it gives the impression that little work has been done on the spectroscopy of sulfur allotropes. However, the vibrational spectroscopy of these allotropes has been the subject of intense investigation for many years (see review by Trofimov et al. *Journal of Sulfur Chemistry*, 30, 518, 2009). Other publications on the vibrational spectroscopy of sulfur allotropes include: Zysman-Colman et al. (*Journal of Sulfur Chemistry*, 29, 309, 2008), Steudel and Schuster (*Journal of Molecular Structure*, 44, 143, 1978), Eckert and Steudel (*Elemental Sulfur and Sulfur-Rich Compounds II*, pp. 31-98, 2003), and the many book chapters famously published by Beat Meyer. These papers should be referenced and this paragraph rephrased.
4. There is a typo in the penultimate line of the second paragraph of the section titled "Destruction Pathways of Charged Sulfur Allotropes" (i.e., he vs. the).
5. On page 3 of the article, it is stated that BOMD simulations were carried out at a temperature of 50 K, which was used as the upper limit for the experimental temperature. It would be useful to state in the text what the experimental temperature was suspected to be, as well as any uncertainty in this value.
6. In the conclusion the authors state that S₈ could be a major sink for sulphur in space, the authors have not mentioned anywhere in the text the possible contribution of larger allotropes. This is particularly important, since some of these allotropes (e.g. the S₁₂ allotrope in particular) are known to also be very stable (see work by Fadyaeva et al. *Physical Chemistry Chemical Physics* 25, 9294, 2023). The authors should discuss the possible roles of large sulphur allotropes in contributing to the sulphur depletion problem, and what are the implications (if any) of their stability on the results of this study?
7. If cross sections have been determined as text suggests could they be provided in an annex with presentation and discussion of evaluated uncertainties?

Reviewer #2 (Remarks to the Author):

Dear Editor,

I have completed my review of manuscript no. NCOMMS-24-09781, titled "Laboratory Infrared Spectra and Fragmentation Chemistry of Sulphur Allotropes" submitted for publication in Nature Communications.

Overall, I found the results presented in the manuscript to be scientifically rigorous and to be of value to the astrochemistry research community. However, before recommending the manuscript for publication, I would like to see a few (relatively minor) changes to some aspects of the text.

1. In the second paragraph of the introduction, the authors state that "volatile sulfur ice reservoirs such as OCS and H₂S can account for at most a few percent of the missing sulfur." However, to the best of my knowledge, H₂S has yet to be detected in interstellar icy mantles (see, e.g., McClure et al. *Nature Astronomy*, 7, 431, 2023). I believe the authors meant to write SO₂ instead of H₂S.

2. Although the authors do a good job of succinctly describing the logic behind the conversion of volatile (i.e., atomic or refractory) sulfur to refractory (i.e., allotropic or mineral sulfur), they do not go into any detail about possible methods by which this may occur. A few short sentences on this really should be added, particularly since great strides in this field of research have been made in the past few years. For instance, Cazaux et al. (*Astronomy and Astrophysics*, 657, A100, 2022) showed that the photo-processing of H₂S ice leads to the formation of refractory sulfur allotropes, while Mifsud et al. (*Frontiers in Chemistry*, 10, 1003163, 2022) showed that electron irradiation of SO₂ or H₂S generates an apparent "depletion" of observable sulphur. Moreover, Coulomb-enhanced freeze out of sulphur ions onto dust grains (see Ruffle et al. *Monthly Notices of the Royal Astronomical Society*, 306, 691, 1999; Umebayashi and Nakano, *Publications of the Astronomical Society of Japan*, 32, 405, 1980) may provide the opportunity for a gradual atom-by-atom build-up of sulphur residues.

3. In the penultimate paragraph of the introduction, the authors state that "To date, the only spectroscopic information on sulfur clusters are the rotational spectra of S₃ and S₄". I feel that this statement is at best misleading since it gives the impression that little work has been done on the spectroscopy of sulfur allotropes. This couldn't be further from the truth, as the vibrational spectroscopy of these allotropes has been the subject of intense investigation for many years (see review by Trofimov et al. *Journal of Sulfur Chemistry*, 30, 518, 2009). Other publications that have reported on the vibrational spectroscopy of sulfur allotropes include: Zysman-Colman et al. (*Journal of Sulfur Chemistry*, 29, 309, 2008), Steudel and Schuster (*Journal of Molecular Structure*, 44, 143, 1978), Eckert and Steudel (*Elemental Sulfur and Sulfur-Rich Compounds II*, pp. 31-98, 2003), and the many book chapters famously published by Beat Meyer. As such, I would strongly suggest editing this sentence and making reference to at least some of these works.

4. There is a typo in the penultimate line of the second paragraph of the section titled "Destruction Pathways of Charged Sulfur Allotropes" (i.e., he vs. the).

5. On page 3 of the article, it is stated that BOMD simulations were carried out at a temperature of 50 K, which was used as the upper limit for the experimental temperature. It would be useful to outright state in the text what the experimental temperature was suspected to be, as well as any uncertainty in this value.

6. Although I largely agree with the conclusion that S₈ could be a major sink for sulphur in space, the authors have not mentioned anywhere in the text the possible contribution of larger allotropes. This is particularly important, since some of these allotropes (I am thinking of the S₁₂ allotrope in particular) are known to also be very stable (see work by Fadyaeva et al. *Physical Chemistry Chemical Physics* 25, 9294, 2023). Could the authors expand a little bit about the possible roles of large sulphur allotropes in contributing to the sulphur depletion problem, and what are the implications (if any) of their stability on the results of this study?

Reviewer #4 (Remarks to the Author):

Dear Dr. Ferrari and co-authors,

I have read your manuscript on sulfur allotropes with great interest, as the "sulfur puzzle" is indeed a long-standing problem in astrochemistry.

Your results on the far-infrared spectrum of S8, S4+, and S4- are certainly very valuable for the community, as is the additional evidence for S8 as a sink of interstellar sulfur in dense clouds. The fact that the obtained results from the laboratory experiments and the DFT calculations match provide compelling evidence for the spectral characterization and the fragmentation routes of S8.

My main concerns with the current version of the manuscript are the following:

- While the manuscript overall is well structured and reads well, it is also very technical. As an astrochemist with a non-laboratory background, it was difficult to follow the detailed methodology presented throughout (which I am sure other laboratory experts will appreciate), and jargon in parts (e.g. ORCA, which is only explained in the SI). Given the somewhat broader audience of this journal compared to, e.g., a laboratory astrophysics journal or even a standard astrophysics journal, maybe the work could be made more easily accessible for a broader audience by including more basic explanations and limiting jargon.
- The manuscript mentions the findings of observational and in situ experimental work from JWST and Rosetta, but does not explain the discrepancies. For instance, Rosetta finds sulfur allotropes predominantly in the forms of S2, S3, and S4, while the authors state that S5 and S6 should be very abundant as fragmentation products of S8.
- The JWST predictions (and statements in the text) do not provide sufficient context as to why sulfur allotrope spectral features have not yet been seen or how much deeper future observations would need to be to identify the predicted spectroscopic patterns and whether that is realistic (the authors state that it would be "challenging", but what does that mean?).

Specific and minor comments:

- Abstract, first part on cosmic sulfur abundance (3rd line): It should be made more clear that the underabundance of sulfur is an observational bias, as the refractories are typically hidden from remote observations.
- Abstract, last sentence: atmospheric abundances of sulfur in exoplanetary atmospheres are mentioned, however, the discussion and conclusions do not include any statements on planetary atmospheres (and environmental conditions). Also, the presence of sulfur allotropes was previously suggested, e.g. from Rosetta (for instance, Calmonte et al. 2016).
- Introduction, second paragraph: Maybe a few recent references on sulfur abundances could be included (e.g., Hily-Blant et al. 2021), also for disks.
- Introduction, last paragraph: Why are the S4+ and S4- experiments done at room temperature (which is far from interstellar conditions)? Is this a valid assumption?
- Introduction, last paragraph: "... enabling us to predict lower abundance limits for their astronomical detection using the James Webb Space Telescope (JWST)". It would be helpful to state how much lower the predictions are compared to limits that were derived already.
- Results, first paragraph: It is argued that atomic S is not present because of its high ionization energy of 10.4 eV, but the laser has 10.5 eV/photon. Why is this not sufficient to ionize S?
- Results, "Because the experiment does not allow to obtain absolute absorption cross-sections, ..." This might be helpful to explain for non-experts.
- Results, "The intrinsic width of the bands will though complicate astronomical observations of free S8." - It would also help the readers if this could be better explained. Why does it complicate astronomical observations?
- Fig. 2a): Why are the increases in S5 and S6 significantly higher than the depletion in S8? Is it because S8 is a lot more abundant to begin with/relative increases?
- Fig. 2b): This figure did not become clear to me. What are the arrows pointing out?
- Results, Destruction pathways of charged sulfur allotropes: Are the room temperature conditions and the He gas valid conditions for transferring the results to an interstellar environment, where the temperature is generally low (5K-100K in a star-forming region) and the dominating collision partner is H2? (The He gas questions also applies to the paragraph "Far-infrared spectroscopy of ionic S4+ and S4- allotropes" where it is stated that the presence of He may act as a heat bath.)

- Same paragraph: "although the S5+ channel is again not the thermo-dynamically favored." Typo, he -> the.
- "We have simulated IR profiles of JWST/MIRI observations, indicating that the direct detection of cold gas phase S8 in molecular clouds is challenging" -> what does challenging mean? is it realistic? Is it most likely not achievable? Does it require an unrealistic amount of observing time? (The SI did not really provide a lot more information on what challenging means.)
- Results, last paragraph: the authors may want to consider presenting the numbers in a small table.
- SI, 3: sulfur is composed by -> of?
- SI, 13: Why are the authors using dust extinctions for the diffuse ISM (which does not include dust growth, and probably not ice mantles either), when they make predictions for observations of a "cold interstellar clouds" (= dense clouds)?

Reviewers #1 and 2:

This paper reports some new and important results on the spectroscopy of sulfur allotropes and examine their fragmentation pathways.

The role of sulfur in astrochemistry is an important one with a major problem being the lack of observations of sulfur in many astronomical regions. The determination of the forms in which sulfur may be found is therefore subject of debate and laboratory research, this is the ‘sulfur depletion puzzle.’ The presented paper focuses on sulfur being tied up in several allotropes, in particular S₈ which may be fragmented to form S_x where x=2,3,4 in both anionic and cationic forms. The paper presents experimental IR spectral measurements and the photodissociation of S₈ revealing a characteristic mode at 21.10 μm, which can be detected by the JWST MIRI instrument, as well as other bands at 41.32 and 53.48 μm. The paper used these results to suggest the presence of S₂ S₃ and S₄ in the comet 67P/Churyumov Gerasimenko arising from fragmentation of S₈. The experimental work is supported by DFT calculations. The methodology is well established and the results well defined and discussed however the Authors should address the following points.

Reply: We thank reviewer #1 for the appreciation of our work.

I have completed my review of manuscript no. NCOMMS-24-09781, titled “Laboratory Infrared Spectra and Fragmentation Chemistry of Sulphur Allotropes” submitted for publication in Nature Communications. Overall, I found the results presented in the manuscript to be scientifically rigorous and to be of value to the astrochemistry research community. However, before recommending the manuscript for publication, I would like to see a few (relatively minor) changes to some aspects of the text.

Reply: We are grateful for the positive comments from reviewer #2 about our work.

1. In the second paragraph of the introduction, the authors state that “volatile sulfur ice reservoirs such as OCS and H₂S can account for at most a few percent of the missing sulfur.” However H₂S has yet to be detected in interstellar icy mantles (see, e.g., McClure et al. Nature Astronomy, 7, 431, 2023). I believe the authors may have meant to write SO₂ instead of H₂S.

Reply: We thank the referee for pointing this out. Indeed, H₂S falls into the category of “not observed”, whereas SO₂ is thought to be “possibly observed”. We have replaced H₂S by SO₂ in the paragraph.

2. Although the authors describe the logic behind the conversion of volatile (i.e., atomic or refractory) sulfur to refractory (i.e., allotropic or mineral sulfur), they do not go into any detail about possible methods by which this may occur. A few short sentences on this should be added. I refer them to the papers of Cazaux et al. (*Astronomy and Astrophysics*, 657, A100, 2022) who showed that the photo-processing of H₂S ice leads to the formation of refractory sulfur allotropes, while Mifsud et al. (*Frontiers in Chemistry*, 10, 1003163, 2022) showed that electron irradiation of SO₂ or H₂S generates an apparent “depletion” of observable sulphur. Moreover, Coulomb-enhanced freeze out of sulphur ions onto dust grains (see Ruffle et al. *Monthly Notices of the Royal Astronomical Society*, 306, 691, 1999; Umebayashi and Nakano, *Publications of the Astronomical Society of Japan*, 32, 405, 1980) may provide the opportunity for a gradual atom-by-atom build-up of sulphur residues.

Reply: This a good suggestion by the referee. We have included a few sentences and corresponding references in the introduction to address this point.

3. In the penultimate paragraph of the introduction, the authors state that “To date, the only spectroscopic information on sulfur clusters are the rotational spectra of S₃ and S₄”. This is rather misleading since it gives the impression that little work has been done on the spectroscopy of sulfur allotropes. However, the vibrational spectroscopy of these allotropes has been the subject of intense investigation for many years (see review by Trofimov et al. *Journal of Sulfur Chemistry*, 30, 518, 2009). Other publications on the vibrational spectroscopy of sulfur allotropes include: Zysman-Colman et al. (*Journal of Sulfur Chemistry*, 29, 309, 2008), Steudel and Schuster (*Journal of Molecular Structure*, 44, 143, 1978), Eckert and Steudel (*Elemental Sulfur and Sulfur-Rich Compounds II*, pp. 31-98, 2003), and the many book chapters famously published by Beat Meyer. These papers should be referenced and this paragraph rephrased.

Reply: We apologize for the poor phrasing. Of course, sulfur allotropes have indeed been the subject of investigation for many years. Our intention was to highlight the need for spectroscopic information in conditions mimicking those found in interstellar environments, so at low temperatures and low densities. We thank the referee for the suggested references, which are now cited in the text. The sentence was modified accordingly.

“While sulfur allotropes have been the subject of active investigation [25, 26, 27], spectroscopic information under astrochemically relevant conditions are scarce, with one example being the rotational spectra of S₃ and S₄ [28].”

4. There is a typo in the penultimate line of the second paragraph of the section titled “Destruction Pathways of Charged Sulfur Allotropes” (i.e., he vs. the).

Reply: We apologize for the typo, which was amended.

5. On page 3 of the article, it is stated that BOMD simulations were carried out at a temperature of 50 K, which was used as the upper limit for the experimental temperature. It would be useful to state in the text what the experimental temperature was suspected to be, as well as any uncertainty in this value.

Reply: From previous studies on the same molecular beam setup, the temperature of the beam is expected to range from 40 to 50 K. The following paragraph was included in the text:

“The temperature in previous molecular beam studies in the same experimental instrument was shown to range from 40 to 50 K [33], so we took 50 K as an upper limit here.”

6. In the conclusion the authors state that S₈ could be a major sink for sulphur in space, the authors have not mentioned anywhere in the text the possible contribution of larger allotropes. This is particularly important, since some of these allotropes (e.g. the S₁₂ allotrope in particular) are known to also be very stable (see work by Fadyaeva et al. Physical Chemistry Chemical Physics 25, 9294, 2023). The authors should discuss the possible roles of large sulphur allotropes in contributing to the sulphur depletion problem, and what are the implications (if any) of their stability on the results of this study?

Reply: The referee is correct that other sulfur allotropes, larger than S₈, have been investigated in the past. However, we do not see any indication that larger allotropes are formed under our experimental conditions, and we can therefore not experimentally assess their stabilities. Nevertheless, although existing astrochemical models only consider the formation of sulfur allotropes up to S₈ (see Ref. [18]), we agree with the referee that mentioning that larger allotropes can also be formed is relevant.

“Moreover, we note that in the past sulfur allotropes larger than S₈ have also been suggested to be stable, but that these were not observed here [30].”

7. If cross sections have been determined as text suggests could they be provided in an annex with presentation and discussion of evaluated uncertainties?

Reply: This is a good idea. In view of a similar suggestion by reviewer #2, a list of the calculated cross sections was included in Table S1 of the SI.

Reviewer #4:

Dear Dr. Ferrari and co-authors, I have read your manuscript on sulfur allotropes with great interest, as the "sulfur puzzle" is indeed a long-standing problem in astrochemistry. Your results on the far-infrared spectrum of S₈, S₄₊, and S₄₋ are certainly very valuable for the community, as is the additional evidence for S₈ as a sink of interstellar sulfur in dense clouds. The fact that the obtained results from the laboratory experiments and the DFT calculations match provide compelling evidence for the spectral characterization and the fragmentation routes of S₈.

Reply: We thank the reviewer for the positive remarks about our work.

My main concerns with the current version of the manuscript are the following:

- While the manuscript overall is well structured and reads well, it is also very technical. As an astrochemist with a non-laboratory background, it was difficult to follow the detailed methodology presented throughout (which I am sure other laboratory experts will appreciate), and jargon in parts (e.g. ORCA, which is only explained in the SI). Given the somewhat broader audience of this journal compared to, e.g., a laboratory astrophysics journal or even a standard astrophysics journal, maybe the work could be made more easily accessible for a broader audience by including more basic explanations and limiting jargon.

Reply: We thank the referee for this remark. Indeed, our intention is to target a broad audience, so we have carefully revised the text in order to make it more accessible to non-experts.

- The manuscript mentions the findings of observational and in situ experimental work from JWST and Rosetta, but does not explain the discrepancies. For instance, Rosetta finds sulfur allotropes predominantly in the forms of S₂, S₃, and S₄, while the authors state that S₅ and S₆ should be very abundant as fragmentation products of S₈.

Reply: Calmonte et al. (MNRAS 462, 253) discuss the detection of S₃ and S₄ in the coma of 67P/Churyumov-Gerasimenko. The range of m/z that the DMFS instrument was sensitive to had been increased in flight to 140 to be able to detect the heavier molecules. That excludes the detection of S₅ and heavier allotropes. Calmonte et al. also state that from the data it is not possible to

establish if the S₃ and S₄ detections were due to fragmentation of larger components. Any fragmentation might also have happened in the instrument. There is circumstantial evidence that the source of the S₃ and S₄ detections were dust particles that released these molecules (or their larger parents). In conclusion, the Rosetta results are inconclusive about the presence of heavier allotropes, so there is no contradiction with the laboratory results that predict S₅ and S₆ as fragmentation products. We have included a brief statement about this point in the text:

“Indeed, S₂, S₃ and S₄ were identified at trace abundances in 67P/Churyumov-Gerasimenko [12], measured with an instrument sensitive towards the small sulfur allotropes. Moreover, S₈ was detected in Ryugu samples [21].”

- The JWST predictions (and statements in the text) do not provide sufficient context as to why sulfur allotrope spectral features have not yet been seen or how much deeper future observations would need to be to identify the predicted spectroscopic patterns and whether that is realistic (the authors state that it would be "challenging", but what does that mean?).

Reply: In the simulations we have used the in-orbit performance of MIRI and have chosen a detection limit of 3σ of S₈ absorption at 21 μm . The in-orbit performance of MIRI shows that a S/N of 100 is achievable for source fluxes of 100 mJy, as a typical noise level at 21 μm for exposure times of 1-2 hours is about 1 mJy. We have assumed a background source at 1 kiloparsec distance emitting as a 20000 K black body and a luminosity of 130.000 solar luminosities, which would correspond to a young OB-type main sequence star. A noise level of 1 mJy and S/N of 100 translate to a 3 per cent absorption band that is minimally required to reach a S/N of 3 of the absorption. This is achieved when the intrinsic width of the feature is similar to the spectral resolution of the MIRI instrument at 21 μm , which we have assumed to be $\lambda/\Delta\lambda = 1000$. We have furthermore assumed that half of the cosmic abundance of sulfur is locked into S₈, based on chemical network models. We have also assumed several intrinsic bandwidths of the S₈ resonance, that are motivated in the text. The detectability of S₈ also depends on the column of interstellar gas and dust in the line of sight. To detect S₈ the optical extinction in magnitudes A_v has to be above about 30 magnitudes to reach a S/N of 3. Very high extinction values (above 90-100) result in a lower continuum flux, pushing the S/N at 21 μm below 100. This discussion was revised in SI, something we have made more explicit in the discussion section of the manuscript:

“We have simulated IR profiles of JWST/MIRI observations, indicating that the direct detection of cold gas phase S₈ in molecular clouds is challenging, based on the required S/N ratio of the 21.1 μm band in the IR profiles, under the assumption that a significant fraction of the sulfur content goes into S₈. More details are provided in the SI.”

Specific and minor comments:

- Abstract, first part on cosmic sulfur abundance (3rd line): It should be made more clear that the underabundance of sulfur is an observational bias, as the refractories are typically hidden from remote observations.

Reply: We have included an additional sentence in the abstract to make this point clearer:

“Astronomical observations in denser regions have so far been able to trace only 1 percent of cosmic sulfur, in the form of gas phase molecules and volatile ices, with the missing sulfur expected to be locked in a currently unidentified form.”

- Abstract, last sentence: atmospheric abundances of sulfur in exoplanetary atmospheres are mentioned, however, the discussion and conclusions do not include any statements on planetary atmospheres (and environmental conditions). Also, the presence of sulfur allotropes was previously suggested, e.g. from Rosetta (for instance, Calmonte et al. 2016).

Reply: This is a good suggestion of the referee. We have included a paragraph discussing the implications our results have about the presence of sulfur in the atmospheres of gas giant exoplanets.

“We note that the presence of sulfur in the atmospheres of gas giant exoplanets is used as evidence for photochemistry in an atmosphere that is enhanced in metals. This assumed tracer role of sulfur depends strongly on a non-volatile nature of the sulfur reservoir in planet forming disks. Our study reveals the fragmentation pathways of S₈, which lead to more volatile sulfur allotropes, and hence would increase the volatility of sulfur in disks. If S₈ is indeed a major reservoir of sulfur in planet forming disks, these would thus imply that sulfur is less reliable as a tracer of the metal content in gas giant atmospheres.”

- Introduction, second paragraph: Maybe a few recent references on sulfur abundances could be included (e.g., Hily-Blant et al. 2021), also for disks.

Reply: Following this request, we have included more recent references about sulfur abundances in the introduction.

- Introduction, last paragraph: Why are the S₄⁺ and S₄⁻ experiments done at room temperature (which is far from interstellar conditions)? Is this a valid assumption?

Reply: The experiments on ions cannot be performed on our molecular beam setup, which is designed for investigating neutral molecules. Instead, ions can be studied in an ion trap, which unfortunately cannot operate at low temperatures. Thus, this is purely a technical reason. While the temperatures in interstellar conditions are certainly different from room temperature, the main effect is broader spectroscopic bands, as discussed in the manuscript, but not their frequencies or relative intensities. The same ion trap setup, for example, has been used in the past to investigate fullerene-derivatives, showing good correspondence with astronomical observations. We have added a brief sentence about this point in the Methods section.

“Details of the experimental setup can be found in Ref. [37]; it has been used in the past to record infrared spectra of astrochemically relevant molecules, such as fullerene-derivatives, showing good correspondence with astronomical observation [44].”

- Introduction, last paragraph: "... enabling us to predict lower abundance limits for their astronomical detection using the James Webb Space Telescope (JWST)". It would be helpful to state how much lower the predictions are compared to limits that were derived already.

Reply: The calculations are based on the assumption that half of the cosmic sulfur is in the form of S₈. Currently only 1 to 10 per cent of sulfur is accounted for. Thus, the underlying assumption is that S₈ is the dominant reservoir of sulfur in dense environments. This is made explicit in the discussion section, with a reference to the SI for further details.

- Results, first paragraph: It is argued that atomic S is not present because of its high ionization energy of 10.4 eV, but the laser has 10.5 eV/photon. Why is this not sufficient to ionize S?

Reply: This is a fair point. The ionization energy is the lowest energy at which ionization can be induced, but typically, the ionization efficiency at the ionization energy threshold is low, and increases with energy. Thus, it is possible that atomic S is not seen with sufficient intensity in the mass spectra because the 10.5 eV/photon is just too close to the ionization energy. Another possibility, however, is that atomic S is not present in our molecular beam. A note on this was included in the text:

“A mass spectrum of the typically generated distribution (Supporting Information, Fig. S2) recorded after ionization with 118 nm laser light (10.5 eV/photon) shows allotropes from S₂ to S₈. The sulfur atom is not observed, possibly due to its high ionization energy (10.4 eV), barely below the

energy of the ionization light. The possibility that atomic S is present in the molecular beam, however, cannot be excluded.”

- Results, "Because the experiment does not allow to obtain absolute absorption cross-sections, ..." This might be helpful to explain for non-experts.

Reply: The main reason is that in our experiments the precise number of photons absorbed per molecule is unknown, but it must be higher than one to be able to induce fragmentation. Absolute cross-sections can only be calculated for single-photon absorption. As suggested by the referee, a sentence about this was included to the text:

“Because this experiment relies on the absorption of more than one photon to achieve fragmentation (with the condition that the energy carried by the photons exceeds the fragmentation energy) and we cannot establish how many photons are absorbed per infrared pulse, the experiment does not allow to obtain *absolute* absorption cross-sections. Nevertheless, the *relative* absorption cross-sections are crucial to benchmark computational methods.”

- Results, "The intrinsic width of the bands will though complicate astronomical observations of free S₈." - It would also help the readers if this could be better explained. Why does it complicate astronomical observations?

Reply: For an optimum detection efficiency, the intrinsic bandwidth of the S₈ resonance at 21 μm should be equal or lower than the spectral resolution of the MIRI instrument. For larger bandwidths, the absorption per instrumental spectral resolution element decreases and so a higher S/N would be required to detect the signal. For very narrow line widths, the spectral sampling of the instrument may prevent detection of the signal. Since this point is explained further in the SI, we have referred to it in the phrase mentioned by the referee.

- Fig. 2a): Why are the increases in S₅ and S₆ significantly higher than the depletion in S₈? Is it because S₈ is a lot more abundant to begin with/relative increases?

Reply: Yes, this is exactly the reason. Figure 2a shows the ratio of the intensities with and without IR irradiation, so a small relative signal decrease in S₈, which is much more abundant than the other allotropes, induce large relative signal increases for S₅ and S₆.

- Fig. 2b): This figure did not become clear to me. What are the arrows pointing out?

Reply: We realize that the arrows are not explained in the caption of the figure. They highlight that after dissociation of S_8^+ , its signal decreases while the signal of S_5^+ and S_6^+ increase. We have included the following sentence in the figure caption for clarity:

“The arrows highlight that the signal decrease due to fragmentation of S_8^+ coincides with signal increases of S_5^+ and S_6^+ .”

- Results, Destruction pathways of charged sulfur allotropes: Are the room temperature conditions and the He gas valid conditions for transferring the results to an interstellar environment, where the temperature is generally low (5K-100K in a star-forming region) and the dominating collision partner is H_2 ? (The He gas questions also applies to the paragraph "Far-infrared spectroscopy of ionic S_4^+ and S_4^- allotropes" where it is stated that the presence of He may act as a heat bath.)

Reply: While conditions are different, collision induced dissociation is a well established technique to determine the lowest-energy fragmentation channels of isolated ionic molecules. Therefore, we believe that our findings can be transferred to dissociation induced in interstellar environments, even if the temperature is lower, or if dissociation occurs through collisions with another partner.

- Same paragraph: "although the S_5^+ channel is again not he thermo-dynamically favored." Typo, he -> the.

Reply: The typo was corrected.

- "We have simulated IR profiles of JWST/MIRI observations, indicating that the direct detection of cold gas phase S_8 in molecular clouds is challenging" -> what does challenging mean? is it realistic? Is it most likely not achievable? Does it require an unrealistic amount of observing time? (The SI did not really provide a lot more information on what challenging means.)

Reply: This question was already asked by the same Reviewer, and was answered on Page 6 of this Letter.

- Results, last paragraph: the authors may want to consider presenting the numbers in a small table.

Reply: Following this and a similar comment from reviewer #1, a list of calculated cross sections is presented now in Table S1 of the SI.

- SI, 3: sulfur is composed by -> of?

Reply: The suggested change was applied.

- SI, 13: Why are the authors using dust extinctions for the diffuse ISM (which does not include dust growth, and probably not ice mantles either), when they make predictions for observations of a "cold interstellar clouds" (= dense clouds)?

Reply: The referee makes a good point here, since it is well known that the dust extinction law in molecular clouds differs from that in the diffuse interstellar medium. We have chosen the diffuse interstellar medium extinction law because it is well documented and provides a conservative estimate of the ratio of HI column density to dust extinction, since in molecular clouds grain growth will likely cause the opacity of dust at optical wavelengths to decrease. In addition, the S₈ resonance accessible to JWST is at 21 μm wavelength. At this wavelength there are no strong competing bands of interstellar ices (McClure et al, Boogert et al. 2015). In addition, grains that have grown to a size of 1 to 2 μm are in the Rayleigh limit at that wavelength, reducing the effect of grain growth on the dust extinction at 21 μm. The use of molecular cloud extinction laws implies that S₈ may be detectable at lower values of A_V (in combination with higher values of R_V = A_V/E(B-V)).

REVIEWERS' COMMENTS

Reviewer #3 (Remarks to the Author):

Dear Dr. Ferrari and co-authors,

Thank you for the revised version of the manuscript. The authors have clearly put a lot of effort into addressing the reviewers' comments carefully and comprehensively. In particular, the readers will certainly appreciate the new table with the cross-sections that was added and the expanded references to previous work. Most of the explanations and answers are very compelling, so I only have a few minor points left:

Regarding the Rosetta results (e.g., Calmonte et al. 2016), the Rosina instrument suite could in principle measure masses up to around 180 Dalton, so the detection of S5 should in principle be possible, but these high masses were not regularly measured, so there are only few spectra and maybe they were not conclusive.

JWST predictions: While the assumptions are now described in more detail (and observational predictions are obviously not the main goal of this work), it still does not become clear why the proposed environment (observations towards a young OB star at 1 kpc) is considered the best environment to search for the sulfur allotropes. Maybe the authors could add a short sentence justifying their choice of target.

Minor details:

- Abstract: the new sentence refers to "denser regions". It does not immediately become clear that this refers to molecular clouds (and not a supernova remnant or something else). Maybe it would be worth considering a phrasing like "dense clouds" or "molecular clouds" instead.
- "these would thus imply that sulfur is less reliable as a tracer of the metal content in gas giant atmospheres." -> this?

Reviewers #3:

Thank you for the revised version of the manuscript. The authors have clearly put a lot of effort into addressing the reviewers' comments carefully and comprehensively. In particular, the readers will certainly appreciate the new table with the cross-sections that was added and the expanded references to previous work. Most of the explanations and answers are very compelling, so I only have a few minor points left.

Reply: We thank the reviewer for the appreciation of our work during the revision phase.

Regarding the Rosetta results (e.g., Calmonte et al. 2016), the Rosina instrument suite could in principle measure masses up to around 180 Dalton, so the detection of S5 should in principle be possible, but these high masses were not regularly measured, so there are only few spectra and maybe they were not conclusive.

Reply: We thank the referee for this observation. It would be very interesting to have dedicated measurements focusing on the mass of S₅. However, given that further discussions of the Rosetta results are beyond the scope of our work, we prefer not to further elaborate on this in the manuscript.

JWST predictions: While the assumptions are now described in more detail (and observational predictions are obviously not the main goal of this work), it still does not become clear why the proposed environment (observations towards a young OB star at 1 kpc) is considered the best environment to search for the sulfur allotropes. Maybe the authors could add a short sentence justifying their choice of target.

Reply: Because sulfur is mostly depleted on dense clouds, in order to see the signal from S₈ in absorption, we need a bright light source in the background. In principle, any bright source will suffice in our model, but just as an example, we selected observations towards a young OB star. As suggested by the referee, we included a short sentence about this in the Supplementary Information.

Minor details:

- Abstract: the new sentence refers to "denser regions". It does not immediately become clear that this refers to molecular clouds (and not a supernova remnant or something else). Maybe it would be worth considering a phrasing like "dense clouds" or "molecular clouds" instead.

Reply: The term was change to “dense clouds” as suggested.

- "these would thus imply that sulfur is less reliable as a tracer of the metal content in gas giant atmospheres." -> this?

Reply: The typo was corrected.